# PMMA Application in Piezo Actuation Jet for Dissipating Heat of Electronic Devices

**DOI:** 10.3390/polym13162596

**Published:** 2021-08-05

**Authors:** Yu-Teng Chang, Rong-Tsu Wang, Jung-Chang Wang

**Affiliations:** 1Department of Information Management, Yu Da University of Science and Technology, Miaoli County 36143, Taiwan; 2Department of Marketing and Logistics Management, Yu Da University of Science and Technology, Miaoli County 36143, Taiwan; 3Department of Marine Engineering (DME), National Taiwan Ocean University (NTOU), Keelung 202301, Taiwan

**Keywords:** PMMA, acrylic, actuation jets, PAJ, piezoelectric ceramic, thermal analysis

## Abstract

The present study utilizes an acrylic (PMMA) plate with circular piezoelectric ceramics (PC) as an actuator to design and investigate five different types of piezo actuation jets (PAJs) with operating conditions. The results show that the heat transfer coefficient of a device of PAJ is 200% greater than that of a traditional rotary fan when PAJ is placed at the proper distance of 10 to 20 mm from the heat source, avoiding the suck back of surrounding fluids. The cooling effect of these five PAJs was calculated by employing the thermal analysis method and the convection thermal resistance of the optimal PAJ can be reduced by about 36%, while the voltage frequency, wind speed, and noise were all positively correlated. When the supplied piezoelectric frequency is 300 Hz, the decibel level of the noise is similar to that of a commercial rotary fan. The piezoelectric sheets had one of two diameters of 31 mm or 41 mm depending on the size of the tested PAJs. The power consumption of a single PAJ was less than 10% of that of a rotary fan. Among the five types of PAJ, the optimal one has the characteristics that the diameter of the piezoelectric sheet is 41 mm, the piezoelectric spacing is 2 mm, and the length of the opening is 4 mm. Furthermore, the optimal operating conditions are a voltage frequency of 300 Hz and a placement distance of 20 mm in the present study.

## 1. Introduction

With the advances in integrated circuit packaging technology and the miniaturization process, the density of the circuit packages embedded in electronic components and the speed of such circuits have improved. In addition, the frequency of operation and heat generation have been relatively increased. If electronic components operate under high temperatures over a long period of time, their lifetimes may be shortened and their efficiency may be reduced. The use of piezoelectric sheets with polymeric materials for the purpose of heat dissipation was first proposed by Toda [1,2] in 1978. He presented a piezoelectric fan which was produced by using polyvinylidene fluoride resin 2 (PVF2) to make the fan blades, which were combined with the piezoelectric ceramics to generate a swayable cantilever beam structure. The result was a piezoelectric fan that could be compared with other small fans in terms of their effectiveness when used for heat dissipation in electronic products. In recent years, there has been significant interest conveyed in the polymer of PVDF (PVF2 or polyvinylidene difluoride) due to it revealing the strongest piezoelectric properties among these merchant polymers. PVDF (homo- and co-polymers) is generally synthesized and polymerized under the emulsion or suspension between 5 and 160 °C and between 5 and 350 atm. This contains generating the biaxially aligned film of PVF2 via admixing in changing capacities of polymethyl methacrylate (PMMA), which is miscible with PVDF. PVDF has extreme pureness and a semi-crystal thermoplastic fluoropolymer, which can be used in chemical manufacturing facilities and has high mechanical strength, electronics and electricals, wonderful chemical resistance, specialized fields, good processability, energy-related applications, and piezoelectric and thermoelectric properties [3,4,5,6,7,8]. Chen et al. [9] demonstrated the wonderful performance for practical electrowetting and energy storage applications of the P(VDF-TrFE)/PMMA (PVT/PMMA)-blended films synthesized by a facile solution-blending method. Moreover, the PVT/PMMA blend containing 20 wt% PMMA with significantly enhanced energy storage capability and reduced remnant polarization even turned out to be a superior material for capacitor applications. Mahapatra et al. [10] investigated the latest development in piezoelectric (smart) materials for various applications in energy harvesting and self-powered sensors including vibrations, human motion, and mechanical loads, etc., in detail. They indicated that the enhancing efficiency of light and temperature-based piezoelectric energy harvesting provides endless future possibilities to fulfill growing energy requirements. Furthermore, the technical advancements in materials, device integration, and fabrication procedures in the piezoelectric energy sector will play a major role in dealing with the global energy crisis by exploiting sensors and self-sufficient batteries. Yoo et al. [11,12] employed dual-piezoelectric sheets to improve the vibration behaviors of piezoelectric and polymeric materials. Several types of dual-piezoelectric sheets were designed and then tested with different voltage levels in order to examine the effects of heat exchange by measuring the amplitude of the materials and the wind speed. The findings showed that when the length of a dual-piezoelectric sheet is increased, the harmonic vibration frequency will be reduced. Moreover, when the mezzanine is made from aluminum materials, it can reduce the amount of energy wasted and the cost.

Kalel and Wang [13] studied the integration of a PVDF-based un-bimorph actuator with a layer of one-way shape memory polymers (1W-SMPs) to achieve maximum bending of the piezoelectric cantilever actuator. The results exhibit a maximum bending angle of 40° at a DC field of 20 V/µm after 60 s between the SMP layer mounted at the center of the actuator and a length half of the PVDF layer, in which structure could be utilized for high unidirectional bending piezoelectric microactuators. Wu et al. [14] discussed using the jet airstream of a piezoelectric synthetic jet generator to enforce heat convection effects. A synthetic jet enhances the cooling effect of its components. When the surface temperature reaches 100 °C, the natural power of the convection dissipation is 2 W, whereas when synthetic jet actuators are employed, the maximum dissipation effect can reach 17 W. When the same power is supplied to the actuator, its dissipation power can be three times bigger than that of the common means of dissipating heat. Ko et al. [15] investigated a piezoelectric fan in motionless air utilizing the CFD and observations with an IR camera. The results showed that the temperature of the heat source is regionally lowered by 28 °C and it is very valid to cool it regionally. Ebrahimi et al. [16,17] studied the vortex advancement on all sides of the tips of vibrational cantilever plates with different structures and oscillating behavior. The vortex regimes depended on the length of the cantilever and Reynold number (Re). The results showed that conquering damping can occupy over 50% of the total power input to a piezoelectric fan and exploit it for heat transfer purposes. Smith et al. [18,19] investigated the influence of the Re and the height of the cavity on the synthetic jet flow field. For the same Reynolds number of 2000, as a near flow field of a synthetic jet can generate vortex pairs, it can create a greater jet stream than a continuous jet flow. A distant flow field was similar to a continuous jet flow. Influences of freestream on the piezoelectric fans were carried out to analyze the inlet velocities (from 0 to 7 m/s) at the side of the fan [20]. The influence of the freestream on the tip vortex is major in that the freestream reduces the vortex intensity just after separation from the fan tip. The side vortex is also affected slightly by the freestream. Both vortices are yet deducted by the freestream and drift away from the fan after separation from the fan edge. Wu et al. [21] used piezoelectric sheets in the heat dissipation of electronic components and used the ANSYS software package for experiments and comparison. The simulation results and comparison found few differences. They also discussed the factors that influence the harmonic frequency and harmonic amplitude, including the length of the piezoelectric materials, placement and distribution, and fixed boundary conditions. The analysis was conducted by changing previous parameters to design four different patterns as alternatives in order to determine the optimal construction. Kercher et al. [22] used synthetic jet technology to manage the heat dissipation of electronic components. Their experimental findings indicated that a piezoelectric synthetic jet had two times better cooling efficiency than the fans presently used as cooling technology.

Many research studies have focused on the aerodynamic performance of oscillating piezoelectric fans with flexible beams composed of polymeric materials because of their potential as active cooling mechanisms for thermal management applications. Conway et al. [23] investigated the influence of crossbeam thickness (1 and 3.7 mm) on the flow field generated by an oscillating crossbeam composed of polymeric materials using a custom-designed particle-image velocity (PIV) facility and numerical analysis. The results inform the design for use in thermal management applications and beneficial for thermal applications where there are constricted environments of oscillating cooling solutions. Liu et al. [24] studied the vibration and cooling performances of the piezoelectric cooling fan using finite element and experimental methods. The numerical results have good agreements with the experimental measurements. Additionally, for the cooling purpose, the piezoelectric cooling fan needs to work under the natural frequency. The aspect ratio of the optimal geometry of the fan blade is 2:3. Jalilvand et al. [25] evaluated the cooling performance of various configurations of synthetic jet-based thermal solution modules based on thermal resistance analysis. The results display that more than 12  W of heat can be dissipated by a DCJ (Dual Cooling Jet). Zhou et al. [26] explored and examined an active enhanced impingement cooling of a circular jet using a piezoelectric fan with low power consumption. The results show that the impingement cooling of the jet enhances as the Re increased. Usually, the new jet can supply superior performance of heat transfer at a small gap and a high Re. A circular enclosure divided by a movable barrier has been numerically investigated inside a separated circular attachment in the presence of a flexible wall [27]. The results verify that the degree of distortion of the plate is straightly dependent on the number of forces. A broad scope of PMMA (PolyMethyl MethAcrylate) polymers is regularly employed for various applications in engineering. The distinct properties of PMMA including the low density, cheap, esthetics, suitable physical and mechanical properties, ease of manufacture, and readily melt-processible technique can be fabricated into parts by injection and compression molding. Accordingly, PMMA is frequently applied in chemical processing equipment (e.g., tubes, valves, pumps, pipes, and fittings), sensors, and actuators, etc. Some chemical modifications and mechanical reinforcement techniques involving adding various types of nanofibers, nanoparticles, nanofillers, nanotubes, and hybrid materials (nanoparticles) are made known to enhance the functions of thermal and dielectric properties and tensile strength of PMMA-based materials in recent decades [28,29,30,31,32,33]. The present study utilizes the piezo actuation jet (PAJ) with the PMMA plate for dissipating heat of electronic devices. The polymeric material of PMMA in the present paper is manufactured through a thermoplastic injection molding and rapid-uniform heating and cooling cycle system associated with a vapor chamber technology, which can improve the tensile strength and decrease the deficiency of the welding lines of a plastic product [34,35]. Tsai et al. [36] exhibited that the plastic products with two opposite gates were found to enhance by 6.8 °C and 10 °C of tensile strength compared with the traditional one, and the other plastic product with eight holes plate is reduced from 12 µm to 0.5 µm of the depth of the welding line. 

## 2. Research Methods

A device of piezo actuation jet (PAJ) consists of a circular piezoelectric sheet and acrylic (PMMA) plate, which was a completely new method of heat dispersion in the present paper. The powders of PMMA named ACRYREX ^®®^ CM-207 with high smoothness, clear transparency, and high gloss were supplied by CHI MEI CORPORATION (Taiwan) in this study. This study investigated different design patterns, including layouts with differences in the piezoelectric sheet size, spacing, and area, with the different types of devices ultimately being used with a high-power LED (HI-LED) [37,38,39,40,41,42] to determine the best PAJ used when combining several PAJs in series. Connecting the PAJs in series increases the overall amount of wind and allows for the addition or subtraction of devices according to the area of the heat source to effectively control the volume of the device of the PAJ and achieve the optimal dissipation effect. The research procedure is shown in Figure 1a. The PAJ (piezo actuation jet) device, respectively, includes two strategies containing performance measurement and structural design.

### 2.1. Performance Measurement

Figure 1b consists of a flow chart detailing the performance measurement of PAJ, which contained measurement methods and properties. The individual performance measurement included noise, displacement, and wind speed. These three parameters were employed in a HI-LED module to conduct thermal performance experiments in order to determine the properties of each device through thermal analysis. The properties are the LED thermal properties and thermal resistance.

#### 2.1.1. Noise

A handmade soundproof box was used in the noise experiment as shown in Figure 2, and the rotary fan adopted in the subsequent experiments was used in the sound proof test. The material of the soundproof box sheet is acrylic having a thickness is 5 mm and the size is 350 × 250 × 250 mm^3^, which is covered with wave-shaped soundproof cotton with a thickness of 50 mm. The decibel meter named by DSL-333 (Tecpel Co., Ltd., Taipei, Taiwan) as a measurement range between 30 and 130 dB with a resolution of 0.1 dB and an error of ±1.5 dB and the frequency response is between 30 Hz and 8 kHz. Outside the box, the rotary fan produced a noise with a value of 62.4 dB, while when placed inside the soundproof box, it produced a noise measured at 50.7 dB. The background volume of the testing room was measured to be 41.6 dB, yet after correction according to the standard noise control Equation (1), that value was corrected to 50.1 dB. This proves that the soundproof box has good soundproofing ability. The volume correction of the background was displayed in Equation (1).
(1)$$L_{0}=10log\left(10^{0.1L_{1}}-10^{0.1L_{2}}\right)$$

*L*_0_: Measured value of the intended sound source 

*L*_1_: Measured value of the total volume

*L*_2_: Measured value of background volume

#### 2.1.2. Displacement

Piezoelectric ceramics (PC) can be used to convert electrical energy into mechanical energy and to transfer energy to the metal foil. The experiment of displacement used a clamp to secure the device of PAJ and sensor head. The high-frequency current flowing through the sensor head coil generates a high-frequency magnetic field, which interferes with the degree of the impedance of the sensor head and is converted into a voltage output. While the voltage signal is being transferred to the digital oscilloscope to generate waveforms, it can be calculated into the displacement or amplitude of the metal sheet as shown in Figure 3. The sensing head named EX-305 (KEYENCE, Co., Ltd., Taipei, Taiwan) has a frequency response of 18 kHz and resolution of 0.4 μm. The distance between the sensing head and piezoelectric patches is 1 mm and the operating temperature is between −10 °C and 60 °C.

#### 2.1.3. Wind Speed

The purpose of this experiment was to determine the optimal placement of the PAJ device and the heat source in order to enhance the heat dissipation effect. In general, the volume of air displaced by a rotary fan is inversely proportional to its placement and the wind speed of the fan will decrease with increasing distance. Moreover, when the fan is too close to the heat source, the jet resistance increases, and the heat dissipation effect may be poorer than natural convection. Thus, the placement distance is an important parameter. A clamp was used to fix the device of PAJ and the distance between it and the hot wire anemometer was changed in increments of 5 mm. Wind speeds were measured at distances of 5, 10, 15, 20, and 25 mm, respectively. Figure 4 exhibits the hot wire anemometer named TES-1341(TES Co., Ltd., Taipei, Taiwan), which can measure wind velocity between 0 and 30 m/s with a resolution of 0.01 m/s and an error of ±3%.

#### 2.1.4. Thermal Resistance Network Analysis

The HI-LED as shown in Figure 5a with dimensions of 36 × 34 × 2.6 mm^3^ and power of 10 W dissipation design used in this study primarily utilizes thermal resistance to evaluate the characteristics of the LED package. Additionally, it can be used to judge the dissipation capacity of a heat sink by observing the level of thermal resistance. That is, a greater thermal resistance indicates a weaker heat dissipation effect. Equation (2) reveals the definition of thermal resistance.
(2)$$R_{T}=\frac{T_{j}-T_{a}}{Q}$$

*R_T_*: Total thermal resistance, *T_j_*: Temperature of the interface, *T_a_*: Temperature of the environment, *Q*: Power consumption 

Five devices of PAJ were fabricated using a variety of parameters, such as the piezoelectric sheet spacing, size, and the opening area, to establish performance test methods and to conduct cooling experiments in hopes that the PAJs can be applied to the heat dissipation of HI-LEDs. According to LED cooling modules, the thermal resistance of an LED can be analyzed in two parts. The first part is the thermal resistance of the heat diffusion (*R_L_*_,1_). The second part is the thermal resistance of the natural convection (*R_a_*_,1_), as shown in Figure 5b. The thermal resistance of the heat diffusion (*R_L_*_,1_): heat is transferred to the substrate by heat. Since the surface area of the heat radiation substrate is greater than that of the heat source, the rate of heat diffusion is affected by the thermal conductivity of the material, such that the diffusion resistance that is generated, which can be calculated according to Equation (3), is defined as the temperature difference between the center of the LED substrate temperature (*T_L_*_,1_) and the average temperature of the substrate interface (*T_M_*_,1_) divided by the total power (*Q_in_*).
(3)$$R_{L,1}=\frac{T_{L,1}-T_{M,1}}{Q_{in}}$$

The thermal resistance of heat convection (*R_a_*_,1_): the energy transmission created by the density difference between the substrate temperature and the air is called natural convection. A fan can also be installed to enhance the convection effect, resulting in a phenomenon called forced convection. Heat is dispersed into the air via convection and this transfer process is called the thermal resistance of heat convection. Equation (4) defines the temperature difference between the average temperature of the substrate interface (*T_M_*_,1_) and the ambient temperature (*T_a_*_,1_) divided by the power (*Q_in_*).
(4)$$R_{a,1}=\frac{T_{M,1}-T_{a,1}}{Q_{in}}$$

To simplify, in the diagram of the network analysis $R_{L,1}+R_{a,1}=R_{T,1}$ of the thermal resistance; $R_{T,1}$ is the total thermal resistance as shown in Equation (5).
(5)$$R_{T,1}=\frac{\left(T_{L,1}-T_{a,1}\right)}{Q_{in}}$$

### 2.2. Structural Design-Two Patterns

The device of PAJ composes of circular piezoelectric sheets and a PMMA plate. The material of the circular piezoelectric sheets is Lead Zirconated Titanite (PZT), which XRD (X-ray Diffraction) patterns reveal the chemical formula of Pb(Zr_0.44_Ti_0.56_)O_3_ as shown in Figure 6. The present study investigated different design patterns, including designs with differences in the piezoelectric sheet size, as well as differences in the spacing and the opening area of the piezoelectric sheet. The tested PAJs were divided into two types according to the design of the jet channels. The first type utilized a linear jet path, as shown in Figure 7a, in which the jet channel outlet has smooth, straight lines and a larger opening area. The second type utilized a flared jet channel, as shown in Figure 7b, the area of the outlet was reduced and the flaring was increased, directing the airflow around the shunt, and increasing the heat dissipation area. The materials of the two jet channels are the PMMA manufactured by the technology of the insert injection molding process with the local heating mechanism of the vapor chamber [34,35,36], which improves the final strength of the product. The product of PMMA strength can be enhanced outstandingly and reduce the defect of the welding lines with a yield rate of up to 100% through this fabricating procedure. The PMMA has a density of 1.16 g/cm^3^, a melting point of 135 °C, a glass temperature of 102 °C, the thermal conductivity of 0.23 W/mk, and tensile strength of 78 MPa in the present paper. Figure 7c exhibited the real PAJ of the second type. These three sets of parameters were used to create five devices with different specifications for comparison. The detailed specifications are shown in Table 1. In addition, the previous literature has indicated that increasing the height of the cavity will directly affect the performance of the jet strength and performance. Therefore, the case5 device served as the control for the case1 device because the two devices are made using piezoelectric sheets with the same diameter and the same opening length, but the spacing between the piezoelectric sheets increased from 2 mm to 3 mm.

## 3. Results and Discussion

This study focused on the characteristics of the device of PAJs with respect to their capacity for heat dissipation when used with a HI-LED. The linear jet path may provide a greater amount of wind, but the wind speed was low. While the linear jet path can effectively reduce the thermal resistance of thermal diffusion, for LED lights more than 3 W, the cooling effect is poor. When the device of the piezo actuation jet is too close to the heat source, the high temperature surrounding the heat source will be returned, causing the chamber temperature to rise and reducing the cooling effect. The flared jet path has a narrower outlet area to enhance the wind speed of the jet stream and can significantly reduce the thermal resistance of thermal convection. Moreover, the flared design on both sides of the jet path can increase the cooling area and slightly reduce the diffusion resistance. We altered the structural designs of the devices, as well as their operating conditions and placement in order to investigate the impacts of such changes on the cooling effect. We chose the optimal design and input conditions and then determined how to improve the performance of a device of piezo actuation jet. The size of the piezoelectric sheet will directly influence the performance of the device of the piezo actuation jet. In the present study, we used 41 mm and 31 mm piezoelectric sheets to conduct experiments under the same conditions. The area of the device of the piezo actuation jet was only 4% of the current commercial rotary fan.

### 3.1. Experiment of Performance Measurement

A piezoelectric sheet is a material that converts electricity into sound. The tone and intensity of the sounds produced depend on the voltage and the frequency of the electricity supplied to the sheet. In this study, the voltage was fixed at the maximum value of 30 V. The measurement range of the frequency was between 50 and 450 Hz. At the same time, we investigated the relationship between voltage frequency and metal sheet displacements.

#### 3.1.1. Noise

This experiment made use of a handmade soundproof box. The background noise of the testing room was 41.6 dB, while inside the soundproof box it was 31.4 dB. The results were plotted according to the size of the piezoelectric sheet used in the given device. Figure 8a shows the results for three test devices named by case1, case2, and case5, in which the diameter of the piezoelectric sheet was 41 mm. As can be seen in Figure 8, the devices with piezoelectric sheets of the same size had a similar trend in terms of the growth of the noise produced. There was a significant increase in the volume at 150 Hz. With increasing frequency, the results of the noise intensity formed a parabola. The red line indicates the decibel level tested, 50.7 dB when the rotary fan was placed in the soundproof box. This was 1 dB lower compared to the device of piezo actuation jet at 300 Hz. Since the opening length was longer for the case1 and case5 devices, those devices had fewer places on which to fix the piezoelectric sheets than the case2 device, such that they caused more noise due to the free vibrations of the metal foil. The spacing of the piezoelectric sheet was larger in the case5 device. Therefore, the vibration noise produced by the device was about 1–2 dB louder than that produced by the case1 device, which is not very different. Figure 8b shows the results for the device of PAJs in which the diameter of the piezoelectric sheet was 31 mm (e.g., the case3 and case4 devices). As shown in the figure, the noise produced by the case3 device exhibited linear growth, while the slope of the parabola for the case4 device was roughly similar to a straight line. When the frequency was set at 125 to 225 Hz, the volume and the tone for both devices remained similar. Meanwhile, the case3 device produced a higher volume than the case4 device since it had a longer length of opening. The experimental results showed that the three devices of piezo actuation jets produced noise similar to that of the rotary fan at 300 Hz. At frequencies from 60 Hz–150 Hz, however, the noises caused by the three devices of PAJs were lower than that of the rotary fan. In addition, a device of piezo actuation jet has no wind shear effect; therefore, it can reduce its noise effectively by adopting a low operating frequency.

#### 3.1.2. Displacement

When constructing a device of PAJ, the number of adhesives used and the fixed probes will be slightly different due to human error. However, it can be speculated from the findings that the same types of piezoelectric sheets have similar vibration behaviors. Figure 9a shows the displacement results for the test PAJs in which the diameter of the piezoelectric sheet was 41 mm. As shown in Figure 8, the amount of displacement exhibited a stair-step pattern, increasing every 50 to 100 Hz. The displacement divided by the frequency was between about 0.1 to 0.2 μm. Figure 9b shows the displacement results for the test PAJs in which the diameter of the piezoelectric sheet was 31 mm. When the frequency was below 280 Hz, there were no significant changes in the amount of displacement. The amount of displacement increased dramatically, however, when the frequency range was 290 to 300 Hz; the ratio of the displacement and the frequency was between 0.1 and 0.2 μm. The two different types of piezoelectric sheets exhibited completely different vibration behaviors in the same measurement range, yet for both types, the ratio of the displacement and the frequency was between 0.1 and 0.2 μm. The amount of displacement can be estimated using the input voltage frequency and the vibration is related to the type of piezoelectric sheet.

#### 3.1.3. Wind Speed

Since the wind speeds of the rotary fan and the piezoelectric fan are not high, if the distance from the heat source is too great, the air jet cannot exchange heat with the heat source and the effect of forced convection cannot function properly. In this section, the wind speed is used to determine the cooling effect and the size of the wind velocity perturbation can be regarded as the indicator for the strength of the turbulence. Figure 10 shows the wind speed of devices at different distances from the hot-wire anemometer. The results show that the wind speed and placement of the rotary fan and devices are inversely proportioned. In case3, the wind speeds measured at 15 mm and 20 mm are larger than at 10 mm. This is because the wind speed is weak in case3 and generates a pair of vortexes at 15 to 20 mm to disturb the ambient air. Therefore, the wind speed is higher. When placed at 25 mm, it will not be affected. In case4, a larger wind speed is generated at 20 mm, while at 25 mm it is not affected. The devices of PAJ are unstable at low frequencies. The interval time between expansion and compression is longer so that it should be no outward expansion flow field, but only the inward airflow into the cavity. When a higher wind speed is generated during the compression, it causes unstable wind speeds. Therefore, accuracy is lower. When frequencies are above 150 Hz, the time interval is shortened and it can be regarded as a continuous outward airflow and the wind speeds will stabilize.

### 3.2. LED Thermal Performance Experiment

For a single LED, the operating temperature is approximately −40 °C to 80 °C, yet the temperature of the light-emitting crystal can reach about 120 °C which has a tremendous impact on the luminous efficiency and the lifespan, so the temperature must be kept under 80 °C. The temperature of the LED module will be stable after 30 min; therefore, it was recorded every 40 min in this experiment. The original design of LED was for 3 W in the present study. However, the natural temperature of the jet exceeds 80 °C at 4 W, so the wattage supply was no longer increased. Since the power supply can not directly set the wattage and only the voltage and current can be altered, these were adjusted to a similar wattage. The energy supplied to the LED will not be completely converted to light; currently, the light efficiency of white LEDs is about 30%, while the other 70% is converted to heat. The devices of PAJ used in the experiment were tested under the same operating conditions; the ambient temperature was 25.3 °C, the input voltage was 30 V, the voltage frequency was 300 Hz, and they were, respectively, placed at distances of 5 mm, 10 mm, 15 mm, 20 mm and 25 mm to analyze the different cooling effects by observing the changes in thermal resistances. Figure 11a compares the position and the total thermal resistance. Among the five designs, case2 has the best cooling effect when placed at 20 mm, reducing the LED module temperature to 36.4 °C, which is lower than the natural convection when supplied 1 W. Therefore, it can continuously increase the wattage of LED lights to provide better illumination capability. The worst cooling effects were achieved when placing the device at 5 mm in case4 and at 25 mm in case3. It can be seen that even though the device of PAJ consisted of small pieces of piezoelectric sheets with a diameter of 31 mm and had a flared jet path and smaller spacing, its cooling effect is not ideal from the present findings. This is mainly because the measured amplitude of displacement in the vibration experiment based on a diameter of 31 mm is small. The volume flow rate of the air that can be forced into the cavity is low and the jet strength is weak so that there was no significant improvement in convection. When the device of PAJ was placed at 5 mm in case5, it lowered the temperature of the LED module to 26.3 °C, whereas case1 reduced the temperature to 27.1 °C. The results show that when increasing the distance between the piezoelectric sheets, the cooling capability of the device of PAJ deteriorates.

Figure 11b compares the position and heat convection coefficient. The heat convection coefficient for case1 is 129 W/m^2^K placed at 5 mm. When the placing distance increased to 25 mm, the heat convection coefficient reduced to 81.7 W/m^2^K. The efficiency of the heat convection decreased as distance increased. The heat convection coefficient of the case2 is 132.9 W/m^2^K placed at 5 mm. In addition, it can upgrade to three times if placed at 20 mm. The heat convection coefficients of case3 and case4 are about 48 to 98 W/m^2^K, respectively. The enhanced heat convection efficiency is not large compared to other devices of piezo actuation jet, yet it still can be applied to the heat source with low wattage. Case5 can enhance its convection efficiency to between 157% and 288% compared to natural convection. The case1, case3, and case5 can be applied to the heat source at a short distance due to the experimental results and previous analysis. Since they have larger opening lengths and areas, they can inhale a wider range of the air jet around with large air volume but low wind speed. If placed too far away from the heat source, then it can not achieve its cooling effect. The PAJ device placed with 5 mm had the best cooling performance among these five locations. Besides, case2 and case4 had the best heat dissipation effect at 20 mm. case2 and case4 have higher wind speeds, increasing distance is conducive to sucking in the surrounding fluid and to avoid hot air is sucked back again. However, the strength of the airflow is limited. When the distance increases to 25 mm, the strength of the airflow will decrease and is not conducive to heat dissipation.

## 4. Conclusions

The present study designed a novel type of piezoelectric fan and investigated operating input conditions such as voltage frequency, placement distance, spacing between piezoelectric sheets, piezoelectric sheet size, and noise produced. The experimental results show that when a device of piezo actuation jet is placed too close to the heat source, the high temperature will suck back the surrounding fluids, causing the fluid chamber temperature to rise and the cooling effect to be reduced. However, the preferred design and operating conditions can fully make use of the device with a better cooling effect than traditional rotary fans. Additionally, the device should be placed 10 to 20 mm from the heat source to ensure that the temperature of the air returned to the chamber will be able to achieve an optimal cooling effect. However, if the device of the PAJ is placed too far away, it will not be able to send wind to the heat source and will not be able to effectively dissipate heat. The device of 41 mm increases the opening area and the spacing of the piezoelectric sheet, the performance is still better than the device consisting of 31 mm piezoelectric sheets. Moreover, the power consumption comparison results show that the power needed for the device of piezo actuation jet to reduce temperature 1 °C is only 10% to 25% of the power needed of a rotary fan and the device of piezo actuation jet has lower costs. Finally, about 25 devices of PAJ can be connected in series while taking up no more space, which can offer above twenty-five times the number of heat convection effects.

## Figures and Tables

**Figure 1 polymers-13-02596-f001:**
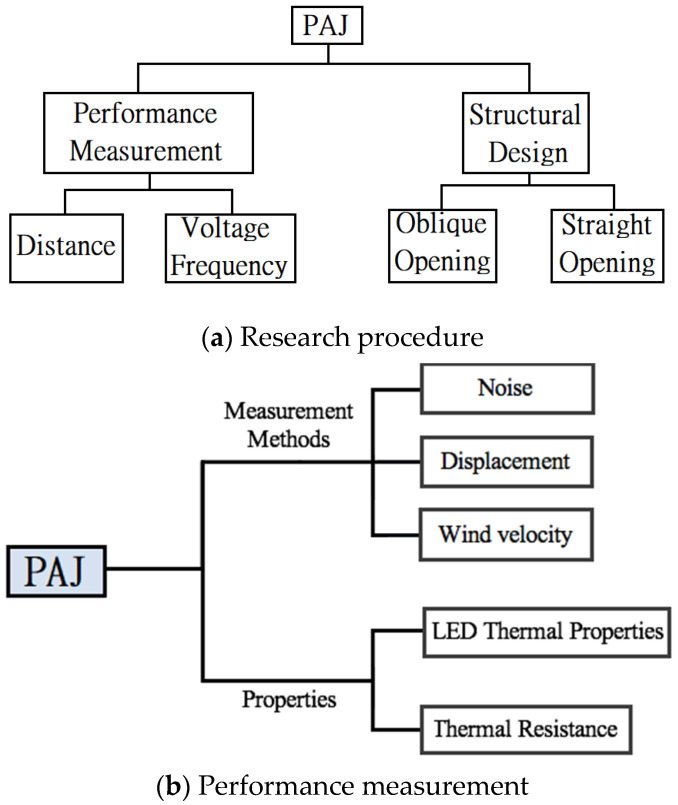
Experimental procedure. (**a**) Research procedure; (**b**) Performance measurement.

**Figure 2 polymers-13-02596-f002:**
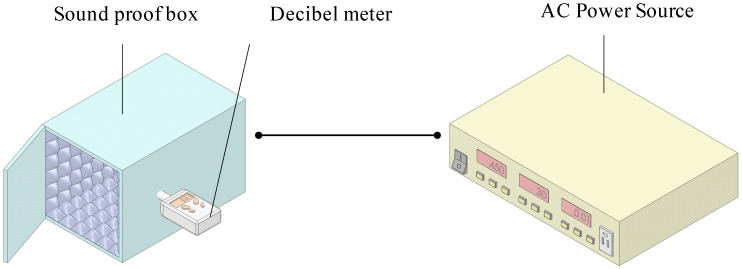
Noise experiment.

**Figure 3 polymers-13-02596-f003:**
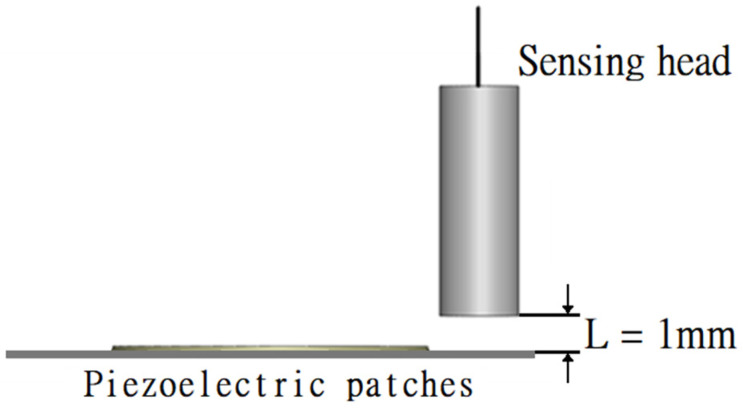
Schematic diagram of jet path.

**Figure 4 polymers-13-02596-f004:**
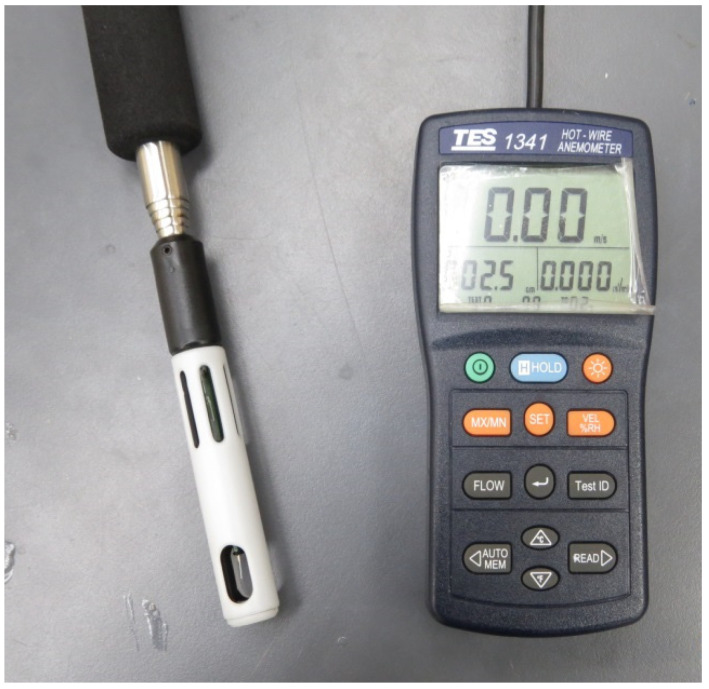
Hot wire anemometer.

**Figure 5 polymers-13-02596-f005:**
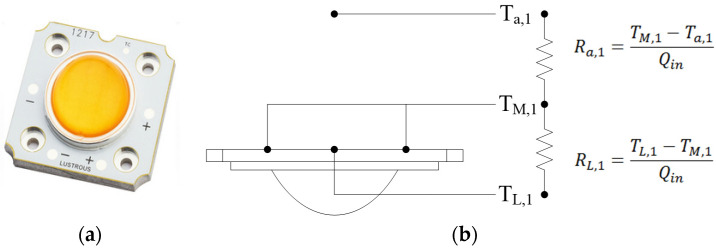
LED Module network. (**a**) HI-LED, (**b**) Thermal resistance network.

**Figure 6 polymers-13-02596-f006:**
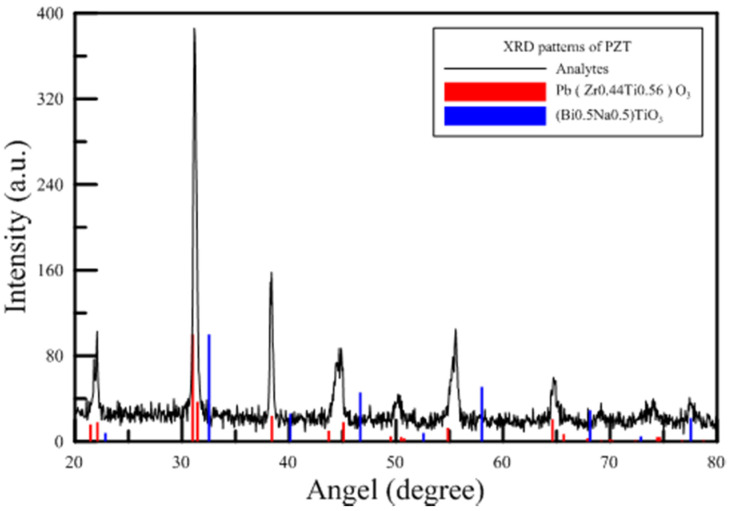
XRD patterns of the present PZT.

**Figure 7 polymers-13-02596-f007:**
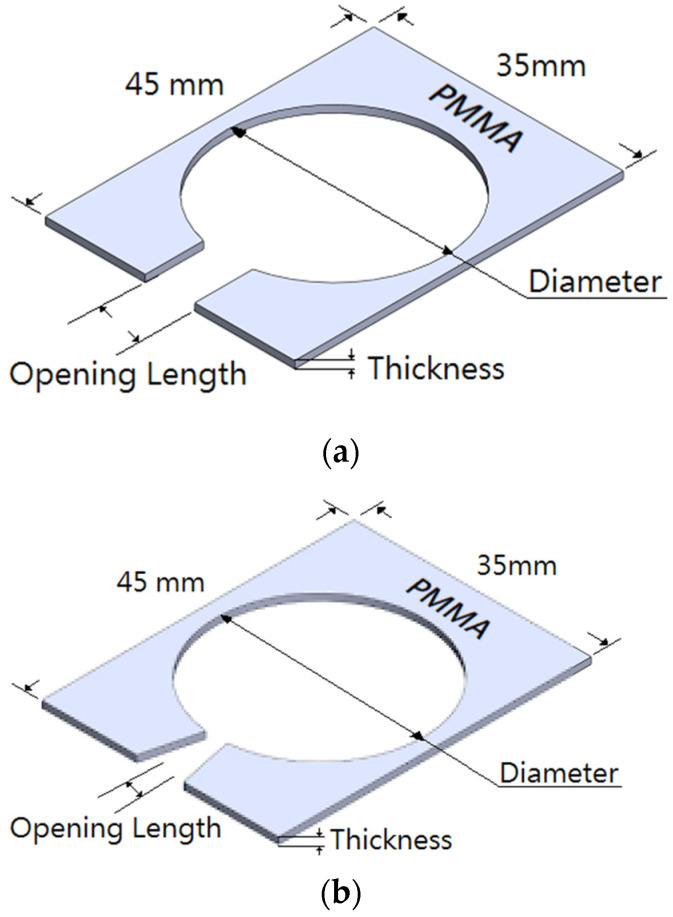
Schematic diagram of jet path. (**a**) Linear type (**b**) Flared type (**c**) Real PAJ.

**Figure 8 polymers-13-02596-f008:**
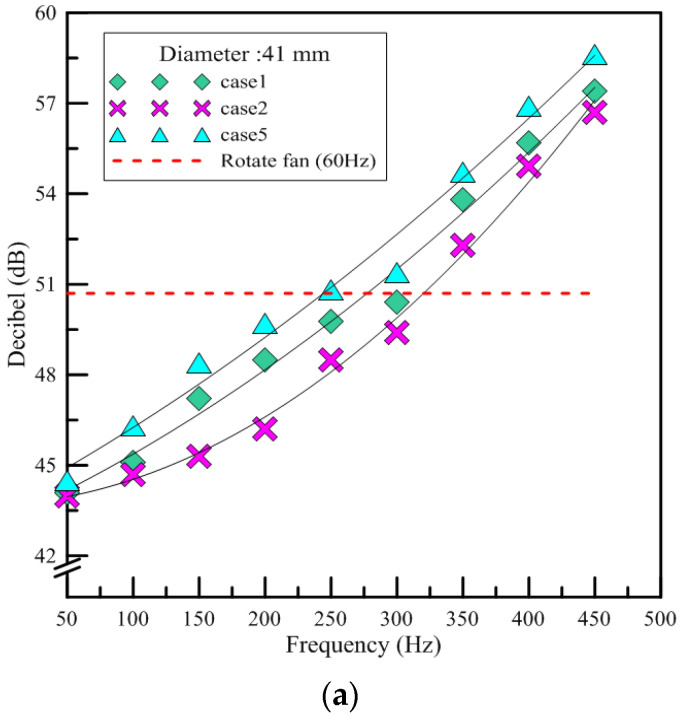
The results of noise measurement. (**a**) Case1, case2, case5 (**b**) Case3, case4.

**Figure 9 polymers-13-02596-f009:**
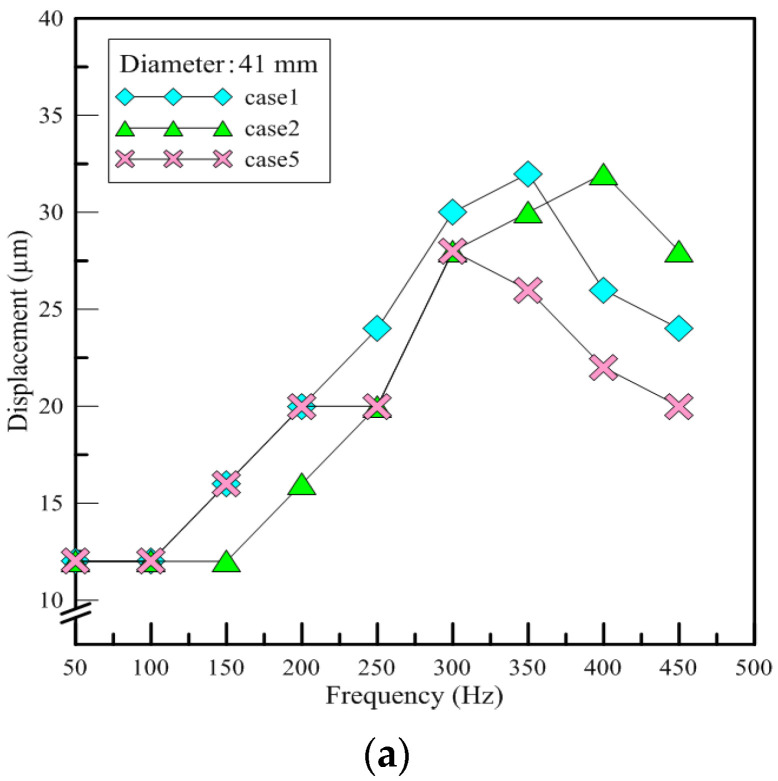
The results of displacement measurement. (**a**) Case1, case2, case5 (**b**) Case3, case4.

**Figure 10 polymers-13-02596-f010:**
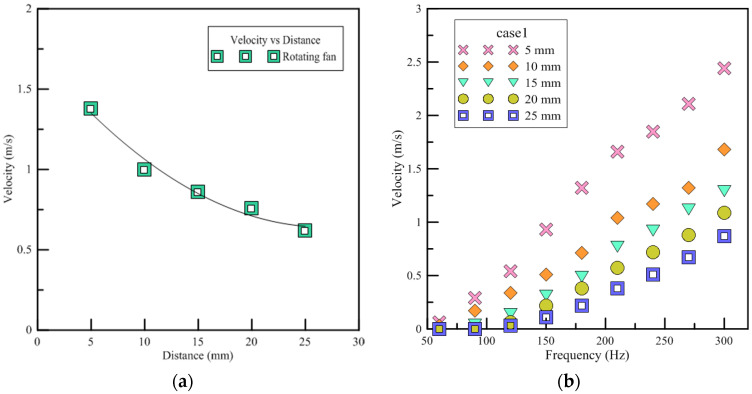
Each device when the wind speed and placement diagrams under different frequencies. (**a**) Rotary fan (**b**) Case1 (**c**) Case2 (**d**) Case3 (**e**) Case4 (**f**) Case5.

**Figure 11 polymers-13-02596-f011:**
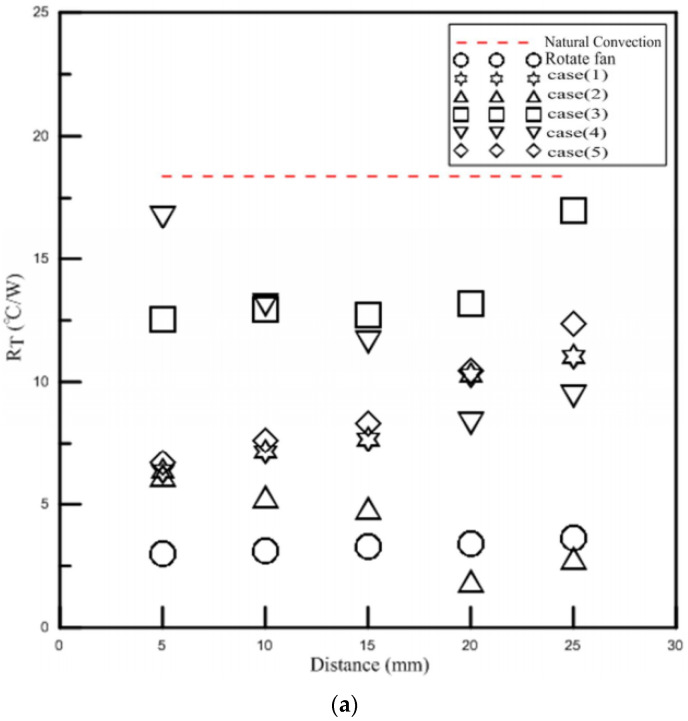
3W LED experimental data. (**a**) Position vs. total thermal resistance (**b**) Position vs. heat convection coefficient.

**Table 1 polymers-13-02596-t001:** Specifications of devices of piezo actuation jet.

Case No.	Diameter (mm)	Cavity Volume (mm^3^)	Opening Length (mm)	Opening Area (mm^2^)	Spacing (mm)
Case1	41	2513.3	10	20	2
Case2	41	2513.3	4	8	2
Case3	31	1413.7	10	20	2
Case4	31	1413.7	4	8	2
Case5	41	3769.9	10	30	3

## Data Availability

All data are offered by the authors for reasonable request and the novel device of piezo actuation jet (PAJ) are available from the authors.

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
