# Peer review of "PMMA Application in Piezo Actuation Jet for Dissipating Heat of Electronic Devices"

_polymers, 2021, doi:10.3390/polym13162596_

Round 1

Reviewer 1 Report

Reviewers' comments:

Manuscript ID: polymers-1305841

Full Title: PMMA Application In Piezo Actuation Jet For Dissipating Heat Of Electronic Devices.

The manuscript describes the PMMA Application In Piezo Actuation Jet For Dissipating Heat of Electronic Devices. The manuscript needs a detailed editing. Some markings are made to just illustrate the extent of editing needed.

The authors need to consider the following comments

- Language needs substantial improvement. Please consult a native English speaker or a language editing service

- In the Abstract: the authors need to improve with more specific short results and conclusions, i.e. academic novelty or technical advantages.

- Keywords – add more related keywords.

- 2.1 Performance measurement - section should be detailed.

- Figure 1 – not clear make clear.

- Authors must but reference for each equation used.

- The quality of Figure 3 is too low.

- 2.6 Thermal resistance analysis - section should be detailed.

- The authors are obliged to repeat the discussion part of the 3.1.2 Displacemeny and 3.1.3 Wind speed.

- Conclusions – too long, should be concise.

- References: there are recent references in 2020-2021 treating the same subject, you can use. And make all references in same format for volume number, page number and journal name.

So that I recommended this manuscript to major revision and for future process.

Author Response

Response to Reviewer 1 Comments

The author is grateful to the reviewers whose comments have helped me improve the manuscript. Thanks Reviewers for their time and effort for the present manuscript. I have reconstructed and reorganized the whole manuscript carefully. And in order to improve the quality of the paper, the whole manuscript has been checked carefully to avoid any grammar or syntax error. Meanwhile, some statements have been rephrased or refined to improve the language usage in the manuscript. I have accounted and discussed the phenomenon more details in the revised version. I have added more comments to explain the physical effects of the plots. The review comments and descriptions of revisions are listed as following. I have included a separate copy of the revised paper in which I highlighted all the modifications made according to all the reviewers’ comments in RED colour. I believe that the present paper is now acceptable for publication.

Point 1: Language needs substantial improvement. Please consult a native English speaker or a language editing service

Response 1: Thanks for the reviewer’s comment. I have revised the whole content and improve the quality of the manuscript carefully. Modifications have been made in the revised manuscript remarked red words.

Point 2: In the Abstract: the authors need to improve with more specific short results and conclusions, i.e. academic novelty or technical advantages.

Response 2: Thanks for the reviewer’s comment. I have been improved the abstract and add some academic novelty or technical advantages. Modifications have been made in the revised manuscript remarked red words.

Point 3: Keywords – add more related keywords.

Response 2: Thanks for the reviewer’s comment. I have been added more related keywords in the Keywords as below.

Keywords: PMMA; Acrylic; Actuation jets; PAJ; Piezoelectric ceramic; Thermal analysis.

Point 4: 2.1 Performance measurement - section should be detailed.

Response 3: Thanks for the reviewer’s comment. Section 2.1 Performance measurement has improved in more detail. Modifications have been made in the revised manuscript remarked red words.

Point 5: Figure 1 – not clear make clear.

Response 4: Thanks for the reviewer’s comment. Fig. 1 has been redrawn to make clear as below.

(a) Research procedure

(b) Performance measurement

Fig.1 Experimental procedure

Point 6: Authors must but reference for each equation used.

Response 5: Thanks for the reviewer’s comment. Modification has been made in the revised manuscript remarked red words.

Point 7: The quality of Figure 3 is too low.

Response 6: Thanks for the reviewer’s comment. Fig. 3 has been redrawn for high quality as below.

(a) Linear type

(b) Flared type

(c) Real PAJ

Fig. 7 Schematic diagram of jet path

Point 8: 2.6 Thermal resistance analysis - section should be detailed.

Response 7: Thanks for the reviewer’s comment. Section 2.6 Thermal resistance analysis has improved in more detail and changed to 2.1.4 Thermal resistance network analysis. Modifications have been made in the revised manuscript remarked red words.

Point 9: The authors are obliged to repeat the discussion part of the 3.1.2 Displacemeny and 3.1.3 Wind speed.

Response 8: Thanks for the reviewer’s comment. Modification has been made in the revised manuscript remarked red words.

Point 10: Conclusions – too long, should be concise.

Response 9: Thanks for the reviewer’s comment. Modification has been made in the revised manuscript remarked red words.

Point 11: References: there are recent references in 2020-2021 treating the same subject, you can use. And make all references in same format for volume number, page number and journal name.

Response 10: Thanks for the reviewer’s comment. References [3-5, 22-28] have been added in the revised manuscript.

Reviewer 2 Report

The manuscript of R.-T. Wang and J.-C. Wang, titled PMMA' Application In Piezo Actuation Jet For Dissipating Heat Of Electronic Devices', describes the use of piezoelectric ceramics (PC) as actuators to design a series of piezo actuation jet analyzing the operating conditions, such as voltage frequency, placement distance, etc. After analyzing five different setups, the author found the optimal conditions.

However, the optimal conditions seem independent of the polymer used (PMMA), being more critical factors the distances between the heat source and the piezo actuation, wind speed, and the piezoelectric sheet size. Control experiments are missing to compare the effects of PMMA (under the same experimental conditions) with, for example, PVF2. The authors do not describe PMMA sources, and the properties of the PMMA used are missing. The title of the manuscript is misleading. PMMA is in the title and is a keyword but is only mentioned once in the text and is unclear where it is in the device.

Others remarks:

  • Piezoelectric ceramics are studying; you should describe which one (Lead zirconate titanate (PZT), barium titanate (BT), and strontium titanate (ST) are the most widely used piezoelectric ceramic materials).
  • A picture of the device will help readers visualize your system and see where the PMMA.
  • In Figure 1, please remove the blue boxes behind the scheme for a better visual.
  • In Figure 2, increase the font size of the legend.

Although the paper is well-organized, the work described is routine, and the manuscript's scope appears better suited for a journal more engineering-oriented.

Author Response

Response to Reviewer 2 Comments

The author is grateful to the reviewers whose comments have helped me improve the manuscript. Thanks Reviewers for their time and effort for the present manuscript. I appreciate his/her comments that my work presents “Although the paper is well-organized”. I have reconstructed and reorganized the whole manuscript carefully. And in order to improve the quality of the paper, the whole manuscript has been checked carefully to avoid any grammar or syntax error. Meanwhile, some statements have been rephrased or refined to improve the language usage in the manuscript. I have accounted and discussed the phenomenon more details in the revised version. I have added more comments to explain the physical effects of the plots. The review comments and descriptions of revisions are listed as following. I have included a separate copy of the revised paper in which I highlighted all the modifications made according to all the reviewers’ comments in RED colour. I believe that the present paper is now acceptable for publication.

Point 1: However, the optimal conditions seem independent of the polymer used (PMMA), being more critical factors the distances between the heat source and the piezo actuation, wind speed, and the piezoelectric sheet size. Control experiments are missing to compare the effects of PMMA (under the same experimental conditions) with, for example, PVF2. The authors do not describe PMMA sources, and the properties of the PMMA used are missing. The title of the manuscript is misleading. PMMA is in the title and is a keyword but is only mentioned once in the text and is unclear where it is in the device.

Response 1: Thanks for the reviewer’s comment. The introduction has been revised and adds more content and references about PMMA and PVF2. Modifications have been made in the revised manuscript remarked red words.

Point 2: Piezoelectric ceramics are studying; you should describe which one (Lead zirconate titanate (PZT), barium titanate (BT), and strontium titanate (ST) are the most widely used piezoelectric ceramic materials).

Response 2: Thanks for the reviewer’s comment. The materials of piezoelectric ceramics are the Lead zirconate titanate (PZT) with Pb(Zr0.44Ti0.56)O3 according to the XRD patterns as below. Modifications have been made in the revised manuscript remarked red words.

Fig. 6 XRD patterns of the present PZT

Point 3: A picture of the device will help readers visualize your system and see where the PMMA.

Response 3: Thanks for the reviewer’s comment. Figure 7 has been added to visualize the present system as below. Modifications have been made in the revised manuscript remarked red words.

(a) Linear type

(b) Flared type

(c) Real PAJ

Fig. 7 Schematic diagram of jet path

Point 4: In Figure 1, please remove the blue boxes behind the scheme for a better visual.

Response 4: Thanks for the reviewer’s comment. Figure 1 has been redrawn as below. Modification has been made in the revised manuscript remarked red words.

(a) Research procedure

(b) Performance measurement

Fig.1 Experimental procedure

Point 5: In Figure 2, increase the font size of the legend.

Response 5: Thanks for the reviewer’s comment. The Fig. 2 has been redrawn as below.

Fig. 3 Schematic diagram of jet path

Reviewer 3 Report

The paper can be accepted after the following corrections:

  1. Construction of proposed piezo-actuation jet has to be presented in more details. Please provide a clear schematic diagram together with the well-explained photography of the prototype.
  2. Method of measurements should be clearly described. Please provide more detailed information about the noise level measurements as well as about thermal resistivity tests.
  3. Uncertainty of measurements should be assessed.
  4. Please develop the conclusions stating practical application of results.

Author Response

Response to Reviewer 3 Comments

The author is grateful to the reviewers whose comments have helped me improve the manuscript. Thanks Reviewers for their time and effort for the present manuscript. I appreciate his/her comments that my work presents “The paper can be accepted after the following corrections:”. I have reconstructed and reorganized the whole manuscript carefully. And in order to improve the quality of the paper, the whole manuscript has been checked carefully to avoid any grammar or syntax error. Meanwhile, some statements have been rephrased or refined to improve the language usage in the manuscript. I have accounted and discussed the phenomenon more details in the revised version. I have added more comments to explain the physical effects of the plots. The review comments and descriptions of revisions are listed as following. I have included a separate copy of the revised paper in which I highlighted all the modifications made according to all the reviewers’ comments in RED colour. I believe that the present paper is now acceptable for publication.

Point 1: Construction of proposed piezo-actuation jet has to be presented in more details. Please provide a clear schematic diagram together with the well-explained photography of the prototype.

Response 1: Thanks for the reviewer’s comment. The construction of the proposed piezo-actuation jet has been presented in more detail. Figure 7 has been added to visualize the present prototype as below. And Modification has been made in the revised manuscript remarked red words.

(a) Linear type

(b) Flared type

(c) Real PAJ

Fig. 7 Schematic diagram of jet path

Point 2: Method of measurements should be clearly described. Please provide more detailed information about the noise level measurements as well as about thermal resistivity tests.

Response 2: Thanks for the reviewer’s comment. The Method of measurements has been described in more detail. Figure 2 has been added to visualize noise level measurements as below. And Modification has been made in the revised manuscript remarked red words.

Fig.2 Noise experiment

Point 3: Uncertainty of measurements should be assessed.

Response 3: Thanks for the reviewer’s comment. And Modification has been made in the revised manuscript remarked red words.

Point 4: Please develop the conclusions stating practical application of results.

Response 4: Thanks for the reviewer’s comment. And Modification has been made in the revised manuscript remarked red words.

Round 2

Reviewer 1 Report

The authors revised the manuscript according to the reviewers' comments.

Author Response

The author is grateful to the reviewers whose comments have helped me improve the revised manuscript. Thanks Reviewers for their time and effort for the present manuscript. I appreciate his/her comments that my work presents “The authors revised the manuscript according to the reviewers' comments.”. I believe that the present paper is now acceptable for publication.

Reviewer 2 Report

The authors have updated the manuscript. However, PMMA has a minimum influence, as is reflected in the conclusions. The PMMA used was prepared by the author, but the properties of the polymer are missing (Mw, DPI, tensile strength). Finally, based on the main scope of this work, it will fit better in MDPI actuators, for example, or a more engineer-oriented journal.  

Author Response

The author is grateful to the reviewers whose comments have helped me improve the revised manuscript. Thanks Reviewers for their time and effort for the present manuscript. I appreciate his/her comments that my work presents “The authors have updated the manuscript.”. I believe that the present paper is now acceptable for publication.

Point 1: However, PMMA has a minimum influence, as is reflected in the conclusions. The PMMA used was prepared by the author, but the properties of the polymer are missing (Mw, DPI, tensile strength). Finally, based on the main scope of this work, it will fit better in MDPI actuators, for example, or a more engineer-oriented journal. 

Response 1: Thanks for the reviewer’s comment. The author “Prof. Dr. Jung-Chang Wang” is the Guest Editor of the Special Issue "Polymer Materials in Sensors, Actuators and Energy Conversion" (https://www.mdpi.com/journal/polymers/special_issues/Polymer_Materials_in_Sensors_Actuators_and_Energy_Conversion) in the "Polymers" journal. About the comments of “based on the main scope of this work, it will fit better in MDPI actuators”, please see the special issue "Polymer Materials in Sensors, Actuators and Energy Conversion" as below and these content of “Polymer-based material applications in sensors, actuators and energy conversion have played a key role in the recently developing areas of smart matter and electronic devices.”, the paper is describing the application of the actuator in electronic devices. I believe that the paper now adapts to the real scope of the work. The content of the Special Issue is below, I believe that the present paper is now acceptable for publication.

Dear Colleagues,

Polymer-based material applications in sensors, actuators and energy conversion have played a key role in the recently developing areas of smart matter and electronic devices. They cover the synthesis, structures, and properties of polymers and composites, including energy harvesting devices and energy storage devices for electro-mechanical (electrical to mechanical energy conversion) and magneto-mechanical (magnetic to mechanical energy conversion), light-emitting devices, and electrical-powered driving sensors. Therefore, modulation of the polymer-based materials and devices for controlling the detection, actuation, and energy of functionalized relative devices is achieved.

Point 2: No modulation of the polymer-based materials to improve the device performance with a minumun description of synthesis, structures, and properties of the polymer (PMMA) have been added to the text. Under the authors logic, this work will also fit to an inorganic chemistry journal, because Pb(Zr0.44Ti0.56)O3 was used.  

Response: Thanks for the reviewer’s comment. The materials of the two jet channels are the PMMA manufactured by the technology of the insert injection molding process with the local heating mechanism of the vapor chamber [31-33], which has an improving impact on the final strength of the product. The product of PMMA strength can be enhanced outstandingly and reduce the defect of the welding lines with a yield rate of up to 100% through this fabricating procedure. The PMMA has a density of 1.16 g/cm3, a melting point of 135 °C, a glass temperature of 102 °C, the thermal conductivity of 0.23 W/mk, and tensile strength of 78 MPa in the present paper. Modifications have been made in the revised manuscript remarked red words. I believe that the present paper is now acceptable for publication.

Reviewer 3 Report

The paper was corrected and can be accepted in the present state.

Author Response

The author is grateful to the reviewers whose comments have helped me improve the revised manuscript. Thanks Reviewers for their time and effort for the present manuscript. I appreciate his/her comments that my work presents “The paper was corrected and can be accepted in the present state.”. I believe that the present paper is now acceptable for publication.
